# Comparison and Optimization of DNA Extraction Methods for Human DNA from Dried Blood Spot Samples

**DOI:** 10.3390/pediatric17020030

**Published:** 2025-03-04

**Authors:** Natalja Van Biesen, Piet Cools, Eline Meyers

**Affiliations:** Department of Diagnostic Sciences, Faculty of Medicine and Health Sciences, Ghent University, 9000 Ghent, Belgium; natalja.vanbiesen@ugent.be (N.V.B.); eline.meyers@ugent.be (E.M.)

**Keywords:** DNA extraction, dried blood spots, quantitative PCR, Chelex, neonatal screening, microsampling, *ACTB*, TREC

## Abstract

Background/Objectives: DNA extraction from dried blood spot (DBS) samples is often applied in neonatal screening programs. Although various methods to extract DNA from DBSs have been described, the optimal approach remains unclear. Therefore, this study aimed to compare and optimize extraction methods to establish a reliable and efficient protocol for human DNA extraction from DBSs. Methods: We conducted a back-to-back comparison of five different DNA extraction methods on 20 DBS samples: three column-based kits (QIAamp DNA mini kit, High Pure PCR Template Preparation kit, DNeasy Blood & Tissue kit) and two in-house boiling methods (one using TE buffer, one using Chelex-100 resin). DNA recovery was measured with DeNovix DS-11 and *ACTB* qPCR. Further optimization of elution volumes and starting material was performed on the best-performing methods (sample size = 5). Additionally, T-cell receptor excision circle (TREC) DNA was assessed by qPCR as an application. Results: The Chelex boiling method yielded significantly (*p* < 0.0001) higher *ACTB* DNA concentrations compared to the other methods. Column-based methods showed low DNA recovery, except for Roche, which showed significantly (*p* < 0.0001) higher DNA concentrations than the other column-based methods, as measured by DeNovix DS-11. Decreasing elution volumes (150 vs. 100 vs. 50 µL) increased *ACTB* DNA concentrations significantly, while increasing starting material (two vs. one 6 mm spot) did not. Conclusions: We identified an easy and cost-effective optimized DNA extraction method using Chelex from DBSs, with an elution volume of 50 µL and 1 × 6 mm DBS punch, which is particularly advantageous for research in low-resource settings and large populations, such as neonatal screening programs.

## 1. Background

The use of blood microsampling, including dried blood spots (DBSs), is increasing in diagnostic fields, drug monitoring, large population studies, and other research contexts. Due to their minimally invasive character, they allow for self-sampling and are cost-efficient [1]. Moreover, dried blood stabilizes analytes, which reduces biohazards and has no special requirements for transport or storage [2,3]. On the other hand, the volume of collected blood is limited (approximately 8.7 µL for a 6 mm disk [4]), and the DBS can show analytical variation [5]. Therefore, depending on the application, DBSs can be an appealing alternative to venous blood.

DNA is one of the multiple targets that can be retrieved from DBSs [3]. DNA extracted from DBSs has valuable clinical applications in neonatal screening programs. For instance, T-cell receptor excision circle (TREC) DNA is quantified from neonatal DBSs in order to screen for severe combined immunodeficiency (SCID). Furthermore, in neonatal screening for spinal muscular atrophy, the SMN1 gene is analyzed from DBS samples [6]. Additionally, in research contexts, it is utilized for applications like methylome profiling [7], next-generation sequencing [8], and detecting circulating tumor DNA [9].

Different methods are described for DNA extraction from DBSs, including two large categories: chemical methods (e.g., column-based silica) and physical methods (e.g., boiling). Column-based silica methods rely on commercial kits, which offer standardized protocols and relatively pure DNA extracts, but are costly and time-consuming. Boiling methods on the other hand are rapid and cost-effective, as they are straightforward techniques where cellular material undergoes degradation due to high temperatures, thereby facilitating the release of DNA from the nucleus and cell [10]. However, the DNA purity is lower since there are no purification steps included [11].

Although numerous methods are described to extract DNA from DBSs, there is no consensus on an optimal method, including for downstream quantification by qPCR [12]. Moreover, studies that conducted back-to-back comparisons of extraction methods on DBSs are limited, especially for human DNA [13]. The few studies that did compare for human DNA overlooked an extraction method using Chelex-100 resin, and tested them for other downstream applications such as whole-genome sequencing (WGS) [8] or methylome profiling [7]. One exception is the study by Simon et al. (2020), which examined and optimized a Chelex-100 extraction method for qPCR on human DNA, but only compared it to one other method using the QIAamp DNA Blood Mini Kit [11].

Therefore, we aimed to compare five methods, including a Chelex-100 resin method, for extracting human DNA from DBS samples and to optimize the most effective methods for downstream qPCR application. The *ACTB* housekeeping gene was targeted to evaluate method performance. Additionally, T cell receptor excision circle (TREC) DNA was also included as a target of interest, given its role as a diagnostic biomarker for SCID in neonates, often assessed using DBSs.

## 2. Materials and Methods

### 2.1. Dried Blood Spot Samples

To compare DNA extraction methods, twenty DBS samples were selected from the SCOPE study, an observational cohort study in nursing home residents and staff [14]. To test whether there was a significant difference in DNA yield between five different DNA extraction kits, a sample size calculation was performed using G*Power 3.1.9.2 a priori, with effect size f = 0.40, α = 0.05, power (1-β) = 0.80, and number of groups = 5, which gave us a sample size requirement of at least 16 DBSs per group. The selection included ten samples collected in February 2021 (stored for approximately three years) and ten in December 2021 (stored for approximately two years). Participants’ capillary blood was collected via finger pricking onto EUROIMMUN Blood Collection Cards (PerkinElmer Health Sciences, Inc., Greenville, SC, USA), which were stored at −20 °C. Since there were multiple spots available per capillary blood sample, paired DBSs were used to compare between methods. Four blank DBS samples per method functioned as negative controls. For the optimization of two selected DNA extraction methods, five participants from Ghent University were recruited and provided twelve spots each, enabling a broader evaluation of alternative protocols. These samples were analyzed within a week after collection. Additionally, one blank DBS per protocol was included as a negative control.

### 2.2. DNA Extraction Methods

Three different column-based kits and two boiling methods were selected based on a literature search, considering their performance, cost, and how commonly they were reported in published studies. Used search terms included “dried blood spot”, “DNA extraction” and “comparison” [10,11,13,15,16,17,18,19,20,21,22]. To allow a comparison across the five different DNA extraction methods, all samples were punched into one 6 mm spot and the elution volumes were set to 150 µL for all five methods.

The QIAamp DNA mini kit (QIAamp) (Qiagen GmbH, Hilden, Germany) was used, following the ‘DNA Purification from Dried Blood Spots’ protocol according to the manufacturer’s instructions [23].

The High Pure PCR Template Preparation kit (Roche) (Roche Diagnostics GmbH, Mannheim, Germany) was selected together with its ‘Isolation of nucleic acids from 25 to 50 mg mammalian tissue’ protocol for the sample lysis and binding [24], as it included a longer incubation step to allow sample release from the filter paper, which takes some time [20].

The DNeasy Blood & Tissue kit (DNeasy) (Qiagen GmbH, Hilden, Germany) was used, following the ‘Purification of Total DNA from Animal Blood or Cells (Spin-Column Protocol)’ of the manufacturer [25]. To release the sample from the DBS punches, they were initially incubated in 180 µL of buffer ATL (tissue lysis buffer) at 85 °C for 10 min, in accordance with other literature using DNeasy [20]. A second modification was the duration of the subsequent incubation step, which was originally 15 min at 56 °C, but was extended to 60 min. The longer incubation step was described in both a literature reference using the DNeasy kit, and the QIAamp protocol for DBSs [20].

A boiling method using Tris-EDTA buffer (TE) was selected, following a protocol described by Bereczky et al. (2005) [15]. Another boiling method using Chelex-100 resin (Chelex) (Sigma-Aldrich, Saint-Louis, MO, USA) was selected, using Chelex-100 resin in sodium form (50–100 mesh-size, dry) [11]. First, a 6 mm DBS punch was incubated overnight at 4 °C in 1 mL of Tween20 solution (0.5% Tween20 prepared in PBS). After the overnight soak, the Tween20 solution was removed and 1 mL of PBS was added to the DBS punch, which was incubated for 30 min at 4 °C. After removing the PBS wash, 50 µL of pre-heated 5% (*m*/*v*) Chelex-100 solution (56 °C) was added to the punch. The mixture was pulse-vortexed for 30 s and then incubated at 95 °C for 15 min, with brief pulse-vortexing every 5 min during the incubation period. The sample was then centrifuged for 3 min at 11,000 rcf to pellet Chelex beads and residual paper. The supernatant was transferred to a new Eppendorf tube using a P200 pipette, and the centrifugation and transfer step were repeated using a P20 pipette for precision. The extracted DNA was stored at −20 °C. The standard operating procedure (SOP) is described in Appendix A.

#### Optimization of Chelex and Roche

For the optimization of selected methods Chelex and Roche, the following elution volumes were compared for both methods: 150 µL, 100 µL, and 50 µL. Additionally, for Chelex with 100 µL elution volume, different starting materials (alternative number of spots and sizes) were compared: one 6 mm punch (‘1 × 6 mm’, total surface of 28.27 mm^2^), two 6 mm punches (‘2 × 6 mm’, total surface of 56.55 mm^2^), and four 3 mm punches (‘4 × 3 mm’, total surface of 28.27 mm^2^).

### 2.3. DNA Quantification Methods

#### 2.3.1. Spectrophotometry Using DeNovix DS-11

DNA extracts were quantified using spectrophotometry via the DeNovix DS-11 (DeNovix Inc., Wilmington, DE, USA). Each DNA extract and negative control was assessed for DNA concentration, and 260/280 ratio (DNA/protein) and 260/230 ratio (DNA/contaminants) as DNA purity parameters. The DNA extracts were measured in triplicate, and the mean concentrations and ratios were calculated. The resulting concentrations served as an indication of the DNA yield. DNA extracts were generally considered ‘pure’ when the 260/280 (DNA/protein) ratio was 1.7–2.0 [26], and the 260/230 (DNA/contaminants) ratio was 2.0–2.2 [26].

#### 2.3.2. ACTB qPCR

The human *ACTB* gene was targeted to evaluate the DNA extraction methods based on the total cell number and DNA quality using an in-house qPCR, performed in duplicate. The qPCR volume was 10 µL, with 1X Probe Mastermix (Roche Diagnostics, Mannheim, Germany), 2 µL of DNA extract (or molecular water, in case of negative template control), 0.5 µM of forward primer, 0.5 µM of reverse primer, and 0.4 µM of probe. The following primer and probe sequences were used: *ACTB* forward: 5′-GGA-TGC-AGA-AGG-AGA-TCA-CTG-3′, *ACTB* reverse: 5′-CGA-TCC-ACA-CGG-AGT-ACT-TG-3′, and *ACTB* probe: 5′-ATTO 425-CCC-TGG-CAC-CCA-GCA-CAA-TG-BHQ^®^-1-3′. The qPCR was run on the LightCycler480 (Roche Diagnostics GmbH, Mannheim, Germany), with an initial denaturation stage (30 s at 95 °C), an amplification stage with 45 cycles (95° for 30 s, 56° for 10 s, and 72° for 30 s), and a cooling stage (5 min at 40 °C). Concentrations were calculated from a standard series using synthetic dsDNA (Integrated DNA Technologies, Coralville, IA, USA) and qPCR results were expressed as log copies/mL.

#### 2.3.3. TREC qPCR

An optimized in-house TREC qPCR was performed in duplicate. The qPCR volume was 10 µL, with 1X Probe Mastermix (Roche Diagnostics, Mannheim, Germany), 2 µL of DNA extract (or molecular water, in case of a negative template control), 0.5 µM of forward primer, 0.5 µM of reverse primer, and 0.125 µM of probe. The primer and probe sequences were TREC forward: 5′-CAC-ATC-CCT-TTC-AAC-CAT-GCT-3′, TREC reverse: 5′-GCC-AGC-TGC-AGG-GTT-TAG-G-3′, and TREC probe: 5′-6-FAM -ACA-CCT-CTG-GTT-TTT-GTA-AAG-GTG-CCC-ACT-TAMRA 3′. The qPCR program ran on the LightCycler480 (Roche Diagnostics GmbH, Mannheim, Germany) and consisted of a polymerase activation stage (2 min at 50 °C), an initial denaturation stage (10 s at 95 °C), an amplification stage with 50 cycles (95° for 10 s, 62° for 30 s, and 72° for 30 s), and a cooling stage (5 min at 40 °C). Concentrations were calculated from a standard series using synthetic dsDNA (Integrated DNA Technologies, Coralville, IA, USA), and qPCR results were expressed as log copies/mL.

### 2.4. DNA Integrity Assessments

To investigate DNA fragmentation for all extraction methods, we analyzed samples from four donors using gel electrophoresis (conditions: 1% agarose gel, run at 140 V for 1 h 30 min, stained with EtBr, and visualized using the Gel Doc XR+ System (Bio-Rad Laboratories, Hercules, CA, USA)).

### 2.5. Data and Statistical Analysis

All qPCR data points represent the mean of duplicate measurements. Means of the DNA extracts were reported with their 95% confidence interval (CI). For the comparison of DNA extraction methods, DeNovix DS-11 concentrations were analyzed with one-way ANOVA (with corrected *p*-values for multiple comparisons), while the purity ratios 260/280 and 260/230 were analyzed with a non-parametric Friedman test, all using QIAamp as the reference due to its protocol specifically designed for DBSs. DNA extracts negative for TREC or *ACTB* were imputed as 1.1 copies/mL. Log-transformed *ACTB* concentrations were evaluated with one-way ANOVA, comparing each method to QIAamp, with corrected *p*-values for multiple comparisons. For the optimization of DNA extraction methods, pairwise comparisons of log-transformed *ACTB* concentrations were conducted using one-way ANOVA with corrected *p*-values for multiple comparisons. While TREC concentrations were also log-transformed, no statistical testing was performed on the TREC data. GraphPad Prism version 10.2.2 (GraphPad Software, Boston, MA, USA) was used for all statistical analyses, and *p*-values below 0.05 were considered significant.

## 3. Results

### 3.1. Comparison of Five DNA Extraction Methods

#### 3.1.1. Total DNA Concentrations and Purity Ratios

DNA concentrations and purity ratios 260/280 and 260/230, measured by DeNovix DS-11, are presented in Figure 1a, Figure 1b, and Figure 1c, respectively. Only four out of five methods are visualized, since the TE DNA extracts were not suitable for spectrophotometry, as debris from dried blood contaminated the buffer and disturbed the transmission of light. Chelex (16.1 ng/µL; CI 14.6−17.6) and Roche (6.78 ng/µL; CI 5.95−7.61) yielded the highest means of total DNA concentration, which were significantly higher compared to QIAamp (1.64 ng/µL; CI 1.42−1.86) (*p* < 0.0001). DNA extraction by DNeasy (2.10 ng/µL; CI 1.67−2.53) yielded similar concentrations as with QIAamp (*p* = 0.108).

DNA extracted with column-based kits had comparable 260/280 purity ratios (*p* > 0.05) near the ideal range, with means of 1.97 (CI 1.49−2.45), 1.62 (CI 1.55−1.69), and 1.72 (CI 1.37–2.07) for QIAamp, Roche, and DNeasy, respectively, while for Chelex, the 260/280 ratio was significantly lower with a mean of 0.97 (CI 0.95−1.03; *p* < 0.0001). For 260/230 purity ratios, all methods were below the ideal, with means of 0.72 (CI 0.33−1.10), 0.56 (CI 0.53−0.59), 0.33 (CI 0.26−0.39), and 0.18 (CI 0.17−0.19) for QIAamp, Roche, DNeasy, and Chelex, respectively.

#### 3.1.2. ACTB Concentrations

The *ACTB* concentrations of DNA extracted by the five different DNA extraction methods are depicted in Figure 2. Notably, the mean *ACTB* concentration of Chelex extracts (6.15 log10 copies/mL; CI 6.05−6.26) was significantly higher compared to that of QIAamp (5.26 log10 copies/mL; CI 5.13−5.39; *p* < 0.0001). Conversely, TE (2.41 log10 copies/mL; CI 1.70−3.11) yielded significantly (*p* < 0.0001) lower *ACTB* concentrations in comparison with QIAamp. The column-based methods Roche (4.98 log10 copies/mL; CI 4.43−5.54; *p* = 0.671) and DNeasy (5.40 log10 copies/mL; CI 5.21−5.58; *p* = 0.090) performed similarly compared to QIAamp. The older samples yielded similar DNA concentrations compared to the more recent samples, suggesting that DBS age may not play an important role in DNA recovery. None of the blank DBS extracts showed *ACTB* amplification.

#### 3.1.3. DNA Integrity

Gel electrophoresis results (Appendix A) showed a very faint smear for Chelex-extracted DNA ranging from 1 to >48.5 kb, suggesting some degree of fragmentation. A faint band for DNeasy-extracted DNA was visible in the range of 15–48.5 kb, indicating the presence of high-molecular-weight DNA at a low concentration.

### 3.2. Optimization of DNA Extraction Methods

#### 3.2.1. Optimization of Elution Volumes

Based on the previous results, Chelex and Roche were selected for further optimization. The *ACTB* concentrations for Chelex and Roche elution volumes 50, 100, and 150 µL are depicted in Figure 3a.

Notably, each of the Chelex elution volumes yielded significantly higher concentrations compared to the corresponding Roche elution volumes. Additionally, higher DNA concentrations were observed when using lower elution volumes. For Chelex, 50 µL (6.41 log10 copies/mL; CI 6.19−6.63) yielded a significantly higher concentration than 150 µL (5.87 log10 copies/mL; CI 5.73−6.00; *p* = 0.008), while no significant difference was observed between Chelex 100 µL (6.24 log10 copies/mL; CI 6.04−6.43) and either 50 µL or 150 µL (*p* = 0.359; *p* = 0.111, respectively). For Roche, *ACTB* yield was significantly higher at 50 µL (5.86 log10 copies/mL; CI 5.79−5.93) compared to 100 µL (5.45 log10 copies/mL; CI 5.40−5.50; *p* = 0.002) and 150 µL (5.32 log10 copies/mL; CI 5.22−5.42; *p* < 0.001) elution volumes, though there was no significant difference between the 150 µL and 100 µL elution volumes (*p* = 0.236). None of the blank DBS extracts showed *ACTB* amplification.

#### 3.2.2. Optimizing Starting Material

Log *ACTB* qPCR concentrations for different starting materials, 1 × 6 mm, 2 × 6 mm and 4 × 3 mm punches, are illustrated in Figure 3b. There were no statistically significant differences in *ACTB* yield across the various starting materials. Starting material 2 × 6 mm (6.46 log10 copies/mL) had a slightly higher mean *ACTB* concentration compared to 1 × 6 mm (6.28 log10 copies/mL; *p* = 0.337) or 4 × 3 mm (6.17 log10 copies/mL; *p* = 0.231). None of the blank DBS extracts showed *ACTB* amplification.

### 3.3. Assessment of TREC Concentrations

As TREC was also a DNA target of interest, we assessed TREC concentrations by qPCR in the DNA extracted by the different methods (Appendix A) and optimizations (Appendix A). However, qPCR showed that TREC concentrations were low to absent in the DNA extracts of our study population. The total mean TREC concentrations for QIAamp, Roche, DNeasy, Chelex, and TE were 0.22 (CI −0.15−0.58), 1.03 (CI 0.20−1.85), 0.79 (CI 0.07−1.51), 0.92 (CI 0.19−1.66), and 0.24 log10 copies/mL (CI -0.17−0.65), respectively, with Roche and Chelex showing the higher concentrations among the methods tested (see Appendix A). Optimization of the elution volume and/or starting material did not substantially increase the TREC yield (see Appendix A). Since TREC results were too low to allow for interpretation, decisions made on selecting and further optimizing the best DNA extraction method were primarily based on the *ACTB* qPCR.

## 4. Discussion

This study aimed to compare five DNA extraction methods for DBSs and further optimize those with better performances. The majority of studies that compare DNA extraction methods for DBSs target pathogen or other non-human DNA [10,13,15,19,20,22]. To the best of our knowledge, among the two studies that compared methods for human DNA, the focus laid on other downstream applications and did not include Chelex as a candidate [7,8]. An exception is Simon et al. (2020), which optimized Chelex for qPCR targeting human DNA, but only compared it to one other method [11].

The Chelex method yielded the highest DNA recovery, at 6 log10 *ACTB* copies/mL, about 10 times more than the column-based kits, which performed consistently at around 5 log10 copies/mL. In contrast, the TE method demonstrated significantly lower efficiency, producing only around 2 log10 *ACTB* copies/mL, which is equivalent to a 1000-fold decrease compared to the column-based methods. The difference in performance between TE and Chelex could be due to the use of Chelex-100 resin, which provides additional protection against DNA degradation, and/or by the presence of Chelex’ washing steps that remove qPCR inhibitors like hemoglobin and IgG antibodies [27], both of which were absent in TE. This hypothesis is supported by the opaque red color of the TE DNA extracts, indicating the presence of lysed red blood cells and inhibitory components like hemoglobin.

Despite the suboptimal 260/280 and 260/230 purity ratios, Chelex still has the highest DNA concentrations as assessed by qPCR. The overall low 260/230 ratios could be attributed to the presence of EDTA [11,23,25] and/or Tris-HCL [23,24,25] in TE buffer, Chelex solution, and elution buffers, as they absorb at 230 nm [28]. Nevertheless, the DeNovix concentrations were relatively low, making qPCR the primary output of interest.

Several studies support our findings, including one by Choi et al. (2014), which reported that boiling methods yielded 60% more DNA from DBS samples than column-based methods (24.5 vs. 14.7 ng/µL) using spectrophotometry [16]. Likewise, Simon et al. (2020) reported that Chelex showed a relatively higher efficiency (54.5%) compared to QIAamp (9.2%) using qPCR [11]. Similarly, Strøm et al. (2014) found that Chelex followed by PCR achieved the lowest limit of detection (LOD) for *Plasmodium* quantification on DBSs (0.16 p/µL) compared to QIAamp (0.5 p/µL) and TE (200 p/µL) [13]. Another boiling method for DNA extraction from DBS samples, developed by Barbi et al. (2000) [29], showed higher cytomegalovirus (CMV) DNA extraction efficacy than column-based methods, as assessed by qPCR [17,19]. In contrast, one study found that QIAamp followed by qPCR had an LOD of 4000 CMV DNA copies/mL, whereas no CMV DNA was detected using Barbi’s method [18].

Based on our results and those of other studies, boiling methods often surpass column-based methods when using DBS samples [13,16,17,19]. A possible explanation could be that boiling temperatures (around 95–97 °C) might effectively release the sample from the DBS and the DNA from the cells. While column-based methods include incubation steps with elevated temperatures (around 50–85 °C), these may not achieve the same level of success in releasing the sample from the filter paper. They are also more dependent on the enzymatic and chemical activity of reagents, which may not be as suitable for sample release from the DBS. This could explain the difference in performances between the more superior Chelex and the column-based methods. However, although the gel electrophoresis results are not fully conclusive, likely due to low DNA concentration, they suggest that Chelex-based extraction yields more fragmented DNA than DNeasy. Nonetheless, Chelex-extracted DNA retains sufficiently large fragments for downstream qPCR. Further research is needed to assess the suitability for long-read sequencing across different extraction methods, although Teyssier et al. (2021) demonstrated its compatibility with WGS [30].

For optimizing the elution volumes, our *ACTB* qPCR results indicate that lower elution volumes using Chelex or Roche yield higher DNA concentrations. It is stated in the Qiagen kits’ manuals that DNA eluting in lower volumes increases DNA concentration [23,25], which we confirmed. In the case of Chelex, the lowest volume possible was 50 µL, since the DBS absorbs some of the fluid.

For optimizing the starting material, we compared 1 × 6 mm (28.27 mm^2^), 2 × 6 mm (56.55 mm^2^), and 4 × 3 mm (28.28 mm^2^). Since the surface areas of 1 × 6 mm and 4 × 3 mm are the same, together with their total captured blood volumes, the only difference is the DBS punch diameter, which did not seem to affect the results. Notably, the use of 2 × 6 mm could theoretically result in higher DNA concentrations due to a higher number of captured cells. In fact, some studies have shown that increased DBS material leads to higher DNA yields, as measured by qPCR or optical density, using boiling and column-based methods [18,21]. We observed a small, non-significant increase in the mean *ACTB* concentration of 2 × 6 mm compared to 1 × 6 mm and 4 × 3 mm, possibly due to the low sample size. This observation aligns with Lee et al. (2024), who compared 1 × 6 mm, 2 × 6 mm, and the whole DBS [12]. To conclude, 1 × 6 mm was selected since the use of a second spot did not result in considerably higher DNA yields.

In addition to *ACTB*, TREC DNA was measured but had very low concentrations, making *ACTB* a better marker to assess performance differences for our DBS samples. This can be easily explained by two things: the low number of cells in peripheral blood that contain TREC, namely recent thymic emigrants [31], and our mature study population, as TREC concentration decreases with age. Concluding, we showed that Chelex is a suited DBS extraction method for human DNA using *ACTB* for downstream qPCR but could not prove the same for TREC due our study population’s demographics.

Other than the efficacy of DNA extraction methods, time efficiency and costs should also be considered. Although Chelex includes an overnight incubation step, the duration of hands-on benchwork is notably shorter compared to the column-based protocols (QIAamp, DNeasy, Roche), especially when working with a higher number of samples. Moreover, the Chelex protocol can be easily scaled up to a 96-well format, allowing high-throughput processing. Chelex is also the cheaper option (estimated cost € < 0.15 per sample) compared to the more expensive column-based kits (for QIAamp approximately €4.30 per sample) [10].

Our findings suggest that Chelex is a promising alternative for use in clinical settings. For instance, national neonatal screening programs that help diagnose SCID rely on quantification of TREC DNA levels, which are naturally high in neonates, from DBSs [32]. They often use column-based methods [32,33], but Chelex offers additional advantages besides performance, including the 96-well format scalability and a lower cost per sample, making it a suitable option. Since our sample selection included DBSs from nursing home residents, which carry similar white blood cell counts (1500/µL around the age of 90 [34,35]) to neonates with lymphopenia (often defined as <2000/µL [36,37]), we believe this DNA extraction method will translate well into the neonatal screening context. Chelex could also be implemented for spinal muscular atrophy screening, where they target the SMN1 gene from DBSs using qPCR [6,38]. Additionally, neonatal screening for conditions like cystic fibrosis, which involves DNA sequencing [8], could possibly benefit from Chelex-based protocols. Moreover, due to the ease of collection, transport, and storage, DBS sampling is increasingly being used in adult populations. Therefore, our findings not only hold value for neonatal screening but also for broader applications where DNA analysis is performed, such as pathogen detection, forensic applications, and (epi)genetic screening [3]. Furthermore, Chelex extraction from DBSs is well-suited for low-resource settings due to its affordability and minimal technological requirements. However, before implementation into clinical or poor resource settings, further validation of the Chelex method for these specific applications would be required.

Our limitations include the low sample size for the optimization, which could have influenced the outcome of the increased starting materials. A second limitation for TREC DNA, specifically, was the inadequate samples, which only had very low TREC counts due to the participants’ demographics. Another limitation is that we did not include an evaluation of other genes with varying lengths or GC contents. Although cell counts and DNA concentration may differ between neonatal blood and (older) adult blood, with neonates typically having nucleated red blood cells while these are absent beyond the neonatal period [39], these differences are not expected to affect the relative performance of the methods. Future studies should include DBS samples from newborns to confirm these findings in the context of newborn screening settings.

## 5. General Conclusions

In this study, we aimed to compare and optimize five human DNA extraction methods from DBS samples for downstream qPCR applications. Using *ACTB* as a marker, the Chelex boiling method yielded DNA approximately 10 times more than column-based kits (QIAamp, DNeasy, Roche) and approximately 10,000 times more than the TE method. Furthermore, lower elution volumes resulted in higher DNA concentrations, while using more starting material did not statistically significantly correlate with higher DNA yields. This led to an optimized DNA extraction method using Chelex, with an elution volume of 50 µL and 1 × 6 mm DBS punch as starting material. Hereby, we found the best performing method to be an easy and cost-effective approach to extract DNA from DBSs, which is particularly advantageous for research in low-resource settings and large-populations, such as neonatal screening programs.

## Figures and Tables

**Figure 1 pediatrrep-17-00030-f001:**
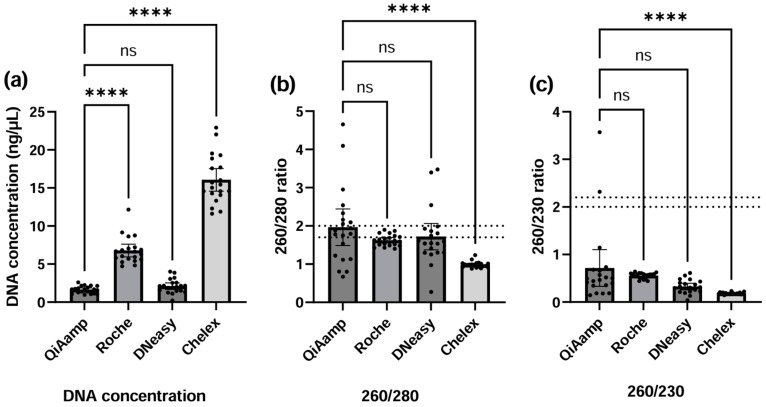
Bar plots (with scatter points) of spectrophotometric measurements of the DNA extracts using DeNovix DS-11. (**a**) The DNA concentrations in ng/µL, (**b**) the 260/280 purity ratio, and (**c**) the 260/230 purity ratio. The means are represented by the height of each bar, and the 95% CI are represented by the error bars. Each dot represents the mean of one DBS sample measured in triplicate. Horizontal dashed lines represent the ranges for 260/280 (1.7–2.0) and 260/230 purity (2.0–2.2) ratios that are generally considered to represent pure DNA. Statistical tests on all three quantities: one-way ANOVA for DNA concentration, and non-parametrical Friedman tests for 260/280 and 260/230, comparing each method against QIAamp as a reference with correction for multiple comparisons. ns = not significant, **** = *p*-value < 0.0001, CI = confidence interval.

**Figure 2 pediatrrep-17-00030-f002:**
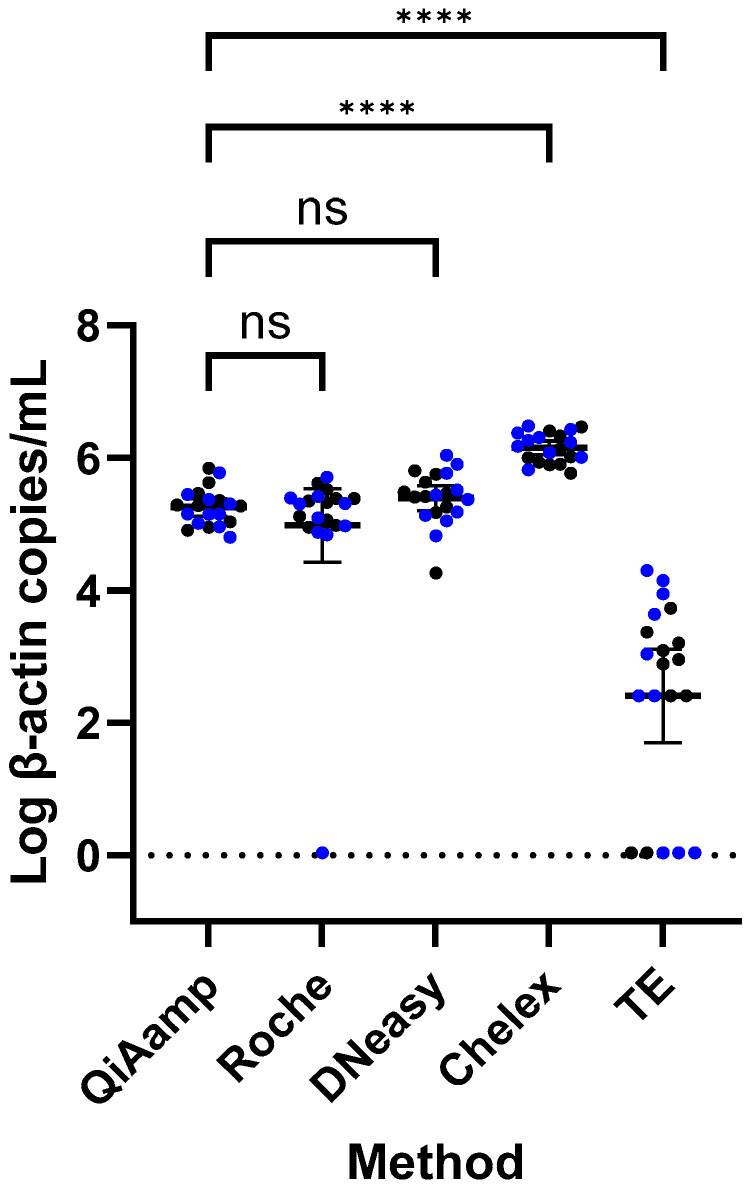
Strip plots with mean and 95% CI of *ACTB* qPCR (log copies/mL) for each of the five DNA extraction methods. The means are represented by the middle horizontal lines, and the 95% CI are represented by the error bars. Each dot represents the mean of the duplicate qPCR measurement (log-transformed). Black dots are samples collected in February 2021 and blue dots are samples collected in December 2021. Statistical testing on DBS samples: one-way ANOVA was used to compare each method to QIAamp as a reference, with correction for multiple comparisons; ns = not significant, **** = *p*-value < 0.0001, CI = confidence interval. Dashed lines on log(*ACTB*) = 0.

**Figure 3 pediatrrep-17-00030-f003:**
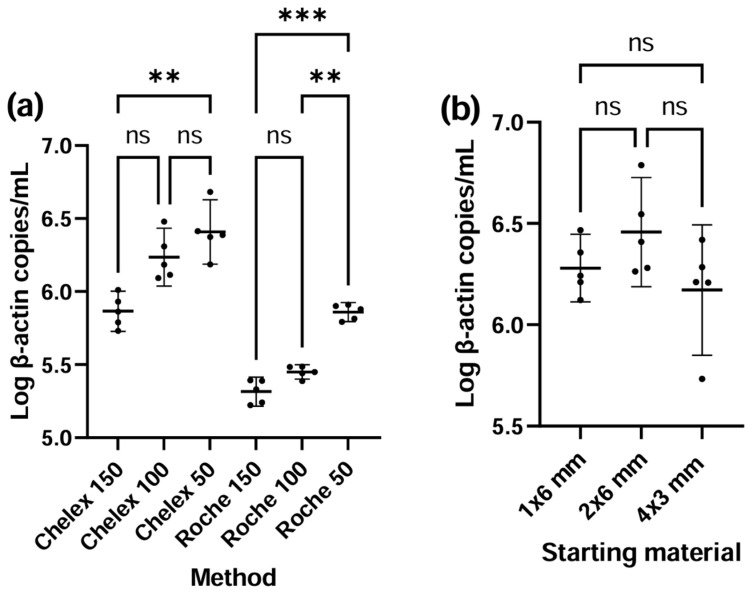
Strip plots with mean and 95% CI of *ACTB* qPCR concentrations (in log copies/mL) for the optimization of (**a**) Chelex and Roche elution volumes and (**b**) Chelex starting material. The means are represented by the middle horizontal lines, and the 95% CI are represented by the error bars. Each dot represents the mean of the duplicate qPCR measurement (log-transformed). Statistical test on *ACTB*: one-way ANOVA to make pairwise comparisons, corrected for multiple comparisons; ns = not significant, ** = *p*-value < 0.01, *** = *p*-value < 0.001, CI = confidence interval.

## Data Availability

Data is available upon request from the corresponding author.

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
