# Peer review of "Comparison and Optimization of DNA Extraction Methods for Human DNA from Dried Blood Spot Samples"

_pediatrrep, 2025, doi:10.3390/pediatric17020030_

Round 1
Reviewer 1 Report
Comments and Suggestions for Authors
This is an important study comparing different methods for obtaining DNA from dried blood spots (DBS). This is highly relevant for Newborn Screening (NBS) Labs, because of the introduction of DNA-based NBS tests into routine programmes (e.g. SCID and SMA). Despite this, the study is narrow - assessing only 5 methods and not including some commonly used protocols. In addition, only 20 DBS were used, and none were from babies. The study does show merit and it feels like an opportunity wasted to not have carried this out more thoroughly. Specific comments below:
1. Why have only 5 methods been evaluated? Some of the most used methods for DNA preparation from blood spots have not been included (e.g. Qiagen Generation solutions, Revvity chemagic kits). This needs to be carried out for the study to hold most value.
2. Why were so few bloods spots used?
3. Newborn screening is carried out on DBS from babies, yet the DBS in this study were from adults. This will affect the results obtained, given the likelihood of differing starting nucleated cell counts in adults versus babies.
4. Many clinical conditions for which DNA extraction from DBS is needed, involve babies with variable white cell counts. No effort was made to include DBS from samples with low counts. This is especially relevant for babies with SCID (one of key reasons for DNA based NBS).
5. DBS were taken at two different timepoints, but it is not clear how old they were when they had DNA extracted. It is well established that DNA recovery from older blood spots is more difficult and would not be used in routine practice. In addition, DBS for NBS are normally stored at room temperature until testing is complete – not at -20°C. Therefore, this study is not relevant for real-world fresh DBS comparison, unless all DNA extractions were carried out within a few days/weeks of being taken. Could the authors clarify?
6. It is mentioned in the text that DNA from blood spots can be used in multiple applications (including WGS). However, some downstream applications require reasonable length genomic DNA as starting material. Do the authors have any qualitative/quantitative assessment on the DNA obtained from each method (e.g. by Tapestation) that will show how fragmented the DNA is? It might be expected that boiling with Chelex leads to more fragmented DNA recovery, inhibiting it’s use in some applications.
Reviewer 2 Report
Comments and Suggestions for Authors
The authors reported the superiority of the DNA extraction method developed in their laboratory (boiling method using Chelex-100 resin). According to the authors, this DNA extraction method was superior to three column-based kits (QIAamp DNA Mini Kit, High Pure PCR Template Preparation Kit, and DNeasy Blood & Tissue Kit) in terms of DNA yield, ease of operation, and reagent economy.
I have been impressed by the authors' determination to create the best methodology possible, rather than relying on commercialized DNA-extraction kits.
However, since this manuscript appears to be a product testing and investigation report by a consumer center, it is highly likely that it would not receive a fair evaluation in peer reviews.
(Major issues)
I think it is completely wrong that the authors put the details of their methodology (boiling method using Chelex-100 resin) in the supplementary information instead of in the main text. The reviewer would like to ask the authors to revise this current version of the manuscript so that your study is properly evaluated.
(Minor Issues)
In this manuscript, β-actin is used as the gene name. I think it should be written in italics.
Reviewer 3 Report
Comments and Suggestions for Authors
This submission offers a valuable contribution by thoroughly comparing various DNA extraction protocols from dried blood spots (DBS). The paper is well-written, methodologically sound, and presents data in a clear and organized manner. I do not have any major objections to the content or approach taken in the study.
Comments:
-
Broader Applicability: While the study emphasizes the relevance of DBS in neonatal screening, it is worth noting that the utility of DBS extends beyond this context. DBS has an increasingly important role in adult health screening and monitoring. Expanding the discussion to highlight this growing utilization among adults would enhance the study's broader applicability and relevance.
-
Inclusion of Statistical Software Details: The manuscript would benefit from a more detailed description of the statistical tools and software used to analyze the data.
- Consider discussing about the cost of the DNA extraction protocols
Round 2
Reviewer 1 Report
Comments and Suggestions for Authors
Thank you for the extra information provided and this will be a useful addition to the scientific field.